# Distribution Learning Meets Graph Structure Sampling

**Arnab Bhattacharyya**
University of Warwick
arnab.bhattacharyya@warwick.ac.uk

**Sutanu Gayen**
IIT Kanpur
sutanugayen@gmail.com

**Philips George John**
CNRS@CREATE & Dept of Computer Science
National University of Singapore
philips.george.john@u.nus.edu

**Sayantan Sen**
Centre for Quantum Technologies
National University of Singapore
sayantan789@gmail.com

**N. V. Vinodchandran**
University of Nebraska-Lincoln
vinod@cse.unl.edu

## Abstract

This work establishes a novel link between the problem of PAC-learning high-dimensional graphical models and the task of (efficient) counting and sampling of graph structures, using an online learning framework. The problem of efficiently counting and sampling graphical structures, such as spanning trees and acyclic orientations, has been a vibrant area of research in algorithms. We show that this rich algorithmic foundation can be leveraged to develop new algorithms for learning high-dimensional graphical models.

We present the first efficient algorithm for (both realizable and agnostic) learning of Bayes nets with a chordal skeleton. In particular, we present an algorithm that, given integers $k, d > 0$, error parameter $\varepsilon > 0$, an undirected chordal graph $G$ on $n$ vertices, and sample access to a distribution $P^*$ on $[k]^n$; (1) returns a Bayes net $\widehat{P}$ with skeleton $G$ and indegree $d$, whose KL-divergence from $P^*$ is at most $\varepsilon$ more than the optimal KL-divergence between $P^*$ and any Bayes net with skeleton $G$ and indegree $d$, (2) uses $\widetilde{O}(n^3 k^{d+1}/\varepsilon^2)$ samples from $P^*$ and runs in time $\mathrm{poly}(n, k, \varepsilon^{-1})$ for constant $d$. Prior results in this spirit were for only for trees ($d = 1$, tree skeleton) via Chow-Liu, and in the realizable setting for polytrees (arbitrary $d$ but tree skeleton). Thus, our result significantly extends the state-of-the-art in learning Bayes net distributions. We also establish new results for learning tree and polytree distributions.

## 1 Introduction

High-dimensional distributions are pivotal in contemporary machine learning, with widespread applications across various domains such as gene regulation networks [46, 16, 17, 40], protein signaling networks [39, 72, 75], brain connectivity networks [53, 78], and psychiatric symptom networks [13, 67, 82]. Probabilistic graphical models provide succinct representations of high-dimensional distributions over an exponentially large sample space such as $\mathbb{R}^n$ or $\{0, 1\}^n$. These models leverage the limited dependence between component variables, encoded by a dependency graph, to describe joint probability distributions over a large set of variables in a succinct and interpretable manner. Probabilistic graphical models such as Bayesian networks, Ising models, and

Gaussian graphical models are extensively utilized to model a wide range of data generation processes in practice (refer to [63, 79, 58] and the references therein). Learning distributions represented by these graphical models is a central challenge with significant theoretical and practical implications.

*The present work focuses on learning an unknown Bayesian network from sample data.* A Bayesian network (Bayes net) with $n$ variables and alphabet size $k$ is a probability distribution over $[k]^n$ defined by a directed acyclic graph (DAG) $G$ on $[n]$. Each node represents a random variable, which is conditionally independent of non-descendants given its parents. By Bayes rule, the distribution factorizes into $n$ conditional probabilities. If $G$ has in-degree at most $d$, it requires at most $nk^{d+1}$ parameters to describe the distribution, significantly reducing the descriptional complexity from $k^n$ parameters required for an arbitrary distribution.

Learning Bayesian network distributions involves two steps: structure learning (identifying the dependency graph) and parameter learning (estimating conditional probability tables). Structure learning methods fall into two categories: constraint-based, which iteratively removes edges by testing for conditional independence, and score-based, which assigns scores to DAGs and frames structure recovery as an optimization problem, often solved using heuristics like greedy hill climbing. The current work broadly fits in the framework of score-based approach. However, instead of optimizing the score directly, we use the framework of *online learning* to reduce the problem to *sampling* from a family of high-dimensional structures.

In the online learning framework, the goal is to design a forecaster that observes a sequence of examples $x^{(1)}, x^{(2)}, \ldots, x^{(T)}$, and at each time (or round) $t$, outputs a prediction $\widehat{p}_t$ based only on $x^{(1)}, \ldots, x^{(t-1)}$. After predicting $\widehat{p}_t$, it observes $x^{(t)}$, and it incurs a loss $\ell(\widehat{p}_t, x^{(t)})$ for a *loss function* $\ell$. The cumulative loss of the forecaster is benchmarked against that of a fixed and known set of *experts*. The goal of the algorithm is to minimize the *regret*, defined as the difference between the cumulative loss over all rounds and the loss it would incur if it were to follow the best expert. Online learning is a well-established field with a wide range of applications in theoretical computer science, including game theory, approximation algorithms, and complexity theory (see [42, 44, 8, 31, 69, 9, 60] and the references therein).

In distribution learning, the experts are all the possible candidate Bayesian networks (up to a sufficient discretization). The observations are random samples from the unknown distribution, and we set the loss function to be the negative log-likelihood $\ell(\widehat{p}, x) = -\log \widehat{p}(x)$. The primary obstacle in applying the online approach to distribution learning lies in ensuring computational efficiency. All the standard forecasting algorithms have running time at least linear in the number of experts. In our case, the experts are all the discretization of all candidate Bayes nets, which is exponentially many. A key insight of our work is the discovery of the close relationship between this computational challenge and the task of efficient counting and sampling of DAGs from a class of DAGs. This link allows us to transfer techniques and algorithms from the counting and sampling literature to the realm of distribution learning, leading to significant new results in learning Bayes net distributions.

## 2 Our Results

We first set up the framework of PAC-learning [76] for distributions; formal definitions appear in Appendix A. We use KL-divergence (denoted as $\mathsf{D_{KL}}$) as the notion of similarity between probability distributions, and we will work with distributions on $[k]^n = \{1, \ldots, k\}^n$. Let $\mathcal{C}$ be a class of such distributions; in our applications, $\mathcal{C}$ will correspond to some family of Bayes nets.

For $\varepsilon > 0$, $A \geq 1$ and two distributions $P$ and $\widehat{P}$ over $[k]^n$, we say $\widehat{P}$ is an $(\varepsilon, A)$-*approximation for* $P$ *with respect to* $\mathcal{C}$ if $\mathsf{D_{KL}}(P\|\widehat{P}) \leq A \cdot \min_{Q \in \mathcal{C}} \mathsf{D_{KL}}(P\|Q) + \varepsilon$. When $A = 1$, we simply say $\widehat{P}$ is an $\varepsilon$-*approximation* of $P$. An algorithm is said to be an *agnostic PAC-learner for* $\mathcal{C}$ if for any $\varepsilon, \delta > 0$ and access to i.i.d. samples from an input distribution $P^*$, it outputs a distribution $\widehat{P}$ which is an $\varepsilon$-approximation for $P^*$ with probability at least $1 - \delta$. If $\widehat{P}$ is not necessarily in $\mathcal{C}$, the algorithm is called *improper*; otherwise, it is called *proper*. Also, the *realizable setting* corresponds to the case when the input $P^*$ is guaranteed to be in $\mathcal{C}$.

It is well-known (e.g., [12]) that given a DAG $G$, the minimum KL divergence between $P^*$ and a Bayes net over $G$ can be written as $J_{P^*} - \sum_{i \in V(G)} I(X_i; X_{\mathsf{pa}_G(i)})$, where $X \sim P^*$, $I$ is the mutual

---

[1]There is a $\mathrm{polylog}(1/\delta)$ dependency here (as opposed to $1/\delta^2$ for the proper learner) hidden in $\tilde{O}(\cdot)$.

|  |  | Chordal graph with indegree $\leq d$ and known skeleton | Tree with unknown skeleton |
|---|---|---|---|
| Realizable | Proper | $\widetilde{O}\left(\max\left\{\frac{n^3}{\varepsilon^2\delta^2}, \frac{nk^{d+1}}{\varepsilon}\right\}\right)$ | $\widetilde{O}\left(\max\left\{\frac{n^3}{\varepsilon^2\delta^2}, \frac{nk^2}{\varepsilon}\right\}\right)$ |
|  | Improper | $\widetilde{O}\left(\frac{nk^{d+1}}{\varepsilon\delta}\right)$ | $\widetilde{O}\left(\frac{nk^2}{\varepsilon\delta}\right)$ |
| Agnostic | Proper | $\widetilde{O}\left(\max\left\{\frac{n^3}{\varepsilon^2\delta^2}, \frac{nk^{d+1}}{\varepsilon}\right\}\right)$ | $\widetilde{O}\left(\max\left\{\frac{n^3}{\varepsilon^2\delta^2}, \frac{nk^2}{\varepsilon}\right\}\right)$ |
|  | Improper | $\widetilde{O}\left(\max\left\{\frac{n^4}{\varepsilon^4}, \frac{nk^{d+1}}{\varepsilon}\right\}\right)^1$ | $\widetilde{O}\left(\max\left\{\frac{n^4}{\varepsilon^4}, \frac{nk^2}{\varepsilon}\right\}\right)$ |

Table 1: Our results: Sample complexities for $(\varepsilon, \delta)$-PAC learning (the $\tilde{O}(\cdot)$ notation hides polylog factors)

information, and $J_{P^*}$ is a constant independent of $G$. Hence, if $\mathcal{C}$ is the class of Bayes nets over DAGs of in-degree $d$, a natural strategy for designing agnostic learning for $\mathcal{C}$ is the following: First approximate the mutual information between any node and any set of $d$ other nodes up to a suitable additive error. Next, maximize the sum of mutual informations between a node and its $d$ parents, over all possible DAGs with in-degree $d$. Iterating over all possible DAG structures would then lead to an algorithm with sample complexity $\widetilde{O}(n^2 k^{d+1}\varepsilon^{-2})$. However, this algorithm has exponential time complexity and the sample complexity is also suboptimal compared to known lower bounds.

In this work, we give an improper agnostic learning algorithm for Bayes nets with indegree $d$ with sample complexity $\widetilde{O}(nk^{d+1}\varepsilon^{-1})$, which is sample-optimal upto polylogarithmic factors. The algorithm is computationally inefficient. Our main contribution is the design of sample and computational-efficient algorithms for new natural classes of Bayes nets, extending the state of the art. In particular, modifying our algorithm for general bounded-indegree Bayes nets, we give computational and time efficient algorithms for learning *chordal-structured Bayes nets with a known skeleton*. Efficient algorithms are currently known only for learning tree-structured distributions ([12, 30, 25]) and for learning polytree-structured distributions with a given skeleton ([24]).

An undirected graph is chordal if every cycle of length four or more has a chord (an edge connecting two non-adjacent vertices in the cycle). Chordal graphs form a significantly broader class than trees and encompass several well-studied graph families, including interval graphs and $k$-trees. Consequently, our results represent a major advancement in the state of the art for learning Bayesian network distributions. Beyond their structural significance, chordal graphs play a crucial role in the study of Bayesian networks particularly in causal Bayesian networks [4, 58]. We describe our results next. The sample complexities of our results are summarized in Table 1.

**Learning with Known Chordal Skeleton**    The *skeleton* of a DAG refers to its underlying undirected graph. We consider Bayes nets having a known *chordal* skeleton with bounded indegree and present an efficient algorithm for learning such distributions.

**Theorem 2.1.** *Let $G$ be an undirected chordal graph on $n$ nodes, and suppose $d$ is a fixed constant. Consider the problem of agnostically learning a distribution w.r.t the class of Bayes nets having skeleton $G$ with indegree $\leq d$. There exist (i) an agnostic improper PAC-learner for this problem using $\widetilde{O}\left(\frac{n^4}{\varepsilon^4} + \frac{nk^{d+1}}{\varepsilon}\right)$ samples that returns an efficiently-samplable mixture of such Bayes nets, and (ii) an agnostic proper PAC-learner using $\widetilde{O}\left(\frac{n^3}{\varepsilon^2\delta^2} + \frac{nk^{d+1}}{\varepsilon}\right)$ samples that returns a single Bayes net. Both algorithms have $\mathrm{poly}(n, k, 1/\varepsilon, 1/\delta)$ running time.*

This is the first result yielding efficient algorithms for agnostic learning Bayes nets on *non-tree* skeletons without further distributional assumptions; see Section 4 for discussion of previous work.

For efficiently learning chordal and polytree distributions, we need to know the correct skeleton (underlying undirected graph). To the best of our knowledge, there is currently no computational hardness result regarding this. Additionally, there have been several works with the known skeleton assumption, even in the context beyond PAC distribution learning. [71] designed an FPT algorithm (in terms of total degree and treewidth) for counting the Markov Equivalence Classes with a given skeleton. On the practical side, several works for Bayes net structure learning first learn a skeleton from the data and then fix the orientations (e.g., see the survey [27], section 4.9.1). However, the

approach of first learning the skeleton and then performing the distribution learning does not have sound theoretical guarantees, since the distance measures in these two contexts are different.

A well-investigated class of Bayes nets is the class of *polytree* distributions: whose DAGs have tree (acyclic) skeletons. Polytrees are especially interesting because they admit fast exact inference [66]. [29] investigated the problem of learning polytree distributions in terms of the negative log-likelihood cost, and showed that it is NP-hard to get a 2-polytree (where indegree is $\leq 2$) whose cost is at most $c$ times that of the optimal 2-polytree for some constant $c > 1$, even if we have oracle access to the true entropies (equivalently, infinite samples). Our distribution learning algorithms in contrast achieve an *additive* approximation in the reverse-KL cost. As a direct corollary of the above theorem, we have the following result for bounded indegree polytrees with known skeleton.

**Corollary 2.2.** *Let $d > 0$ be a fixed constant and $G$ be a given undirected tree. Consider the problem of agnostically learning a distribution w.r.t the class of Bayes nets having skeleton $G$ with indegree $\leq d$, i.e. $d$-polytrees with skeleton $G$. There exist (i) an agnostic improper PAC-learner for this problem using $\widetilde{O}\left(\frac{n^4}{\varepsilon^4} + \frac{nk^{d+1}}{\varepsilon}\right)$ samples that returns an efficiently-samplable mixture of such polytrees, and (ii) an agnostic proper PAC-learner using $\widetilde{O}\left(\frac{n^3}{\varepsilon^2\delta^2} + \frac{nk^{d+1}}{\varepsilon}\right)$ samples that returns a single polytree. Both algorithms have $\mathrm{poly}(n, k, 1/\varepsilon, 1/\delta)$ running time.*

The closest related result is that of [24] who designed a PAC-learner in the realizable setting for polytrees with optimal[2] sample complexity $\widetilde{O}(nk^{d+1}\varepsilon^{-1}\log\delta^{-1})$. However, their analysis crucially uses the realizability assumption, and it was left as an open question in that work to find an efficient agnostic learner for polytrees. The above corollary answers this question.

**Remark 2.3.** We can bound the running time of our learning algorithms for chordal-structured Bayes nets (with known skeleton) as follows: At the outset, for every node $i \in [n]$ and for every choice of the $\leq d$ parents $\mathsf{pa}(i)$, we learn the conditional distribution associated with node $i$ given that choice of parents $\mathsf{pa}(i)$. These are add-one conditional distributions computed from a sufficiently-large ($\widetilde{O}(nk^d/\varepsilon)$) set of samples. Subsequently, the learning algorithm focuses only on the combinatorial problem of learning an acyclic orientation. The running time contribution from the node-distribution learning part is $\widetilde{O}((\Delta k)^d dn^2/\varepsilon)$, where $\Delta$ is the maximum (undirected) degree of the skeleton. Here, $n\binom{\Delta}{d} \leq n\Delta^d$ (for $d \ll n$) bounds the number of all possible (node, parent-set) pairings and $\widetilde{O}(dnk^d/\varepsilon)$ is the time for computing a "good" add-one conditional distribution for a given node and parent-set. Note that the runtime is polynomial even if both $\Delta$ and $d$ are $O(\log n)$. If $d$ is unbounded, then the runtime can grow at an exponential $2^{d\log(\Delta k)}$ rate. Note that, for unbounded $d$, an exponential dependence on the runtime and sample complexity is inevitable since chordal-structured indegree-$(n-1)$ Bayes nets with a fixed skeleton (the complete graph on $[n]$) can capture arbitrary $n$-dimensional distributions (we do not use faithfulness or similar assumptions for distribution learning).

**Learning Tree-structured Distributions**    By *tree-structured distributions* (or simply, trees when the meaning is clear), we mean Bayes nets whose underlying DAG has in-degree 1. They can be equivalently defined as undirected Markov models over (undirected) trees. The celebrated work of [25] developed a polynomial time algorithm for learning tree-structured distributions, if the algorithm is provided the exact mutual information between each pair of variables. PAC-learning guarantees with sample complexity bounds came later [30, 12], In particular, the highlight of these works was establishing that in the realizable setting, i.e., when the samples are being generated by a tree-structured distribution on $[k]^n$, the Chow-Liu algorithm is a PAC-learner with sample complexity $\widetilde{O}(nk^3/\varepsilon)$. While the dependence on $n$ and $\varepsilon$ is tight, it was left as an open question whether a better dependence on $k$ is possible.

Our work answers this in the affirmative:

**Theorem 2.4.** *Let $\mathcal{C}$ be the family of tree-structured distributions over $[k]^n$. There exists an algorithm that for any $\varepsilon > 0$, given sample access to a distribution $P^* \in \mathcal{C}$, returns an $\varepsilon$-approximation $\widehat{P}$ of $P^*$ with probability at least $2/3$ and uses $m = \widetilde{O}(nk^2\varepsilon^{-1})$ samples and $\mathrm{poly}(m)$ running time. The distribution $\widehat{P}$ is a mixture of distributions from $\mathcal{C}$ and is samplable in polynomial time.*

---

[2]Although not stated in the corollary above, in the realizable setting, our techniques also yield sample complexity with the optimal dependence on $n, k$, and $\varepsilon$.

Note that in contrast to Theorem 2.1, here, the algorithm does not know the true skeleton a priori. The output distribution $\widehat{P}$ is a mixture of exponentially many trees but can nevertheless be sampled in polynomial time by using the matrix-tree theorem, as we explain later. We note that the dependence of $k^2$ on the sample complexity is tight. This follows from [14, Theorem 13] (see also [33]), which proves that learning a Bayes net with in-degree $d$ requires $\Omega(nk^{d+1})$ samples, and for tree Bayes nets, the in-degree being $d = 1$, requiring $\Omega(nk^2)$ samples. Learning with mixtures of trees has been studied before ([64, 61, 3, 70]) but in other contexts.

We also note that going beyond trees, the same approach allows us to develop polynomial sample and time algorithms for learning Bayes nets on an unknown DAG whose moralization is promised to have constant vertex cover size. Here, instead of sampling using the matrix-tree theorem, we can utilize a recent result of [50] to sample such DAGs. Details appear in Appendix F.

**Why KL divergence?**  We briefly discuss why learning in KL divergence is relevant. The study of agnostic learning of distributions in KL divergence goes back to at least three decades ago by the works of [2] and [28, 29]. The authors argued that given random samples from an unknown distribution $P$, minimizing the KL divergence is the same as maximizing the log-likelihood in expectation, due to the following equation: $\mathsf{D}_{\mathsf{KL}}(P||Q) = -H(P) - \mathbb{E}_{x \sim P}[\log Q(x)]$, where $H(P)$ is the entropy of $P$. KL divergence also appears in the study of density estimation such as Yang-Barron's construction and covering complexity ([84, 81]). [56] also studied the complexity of distribution learning in terms of KL divergence and gave a coding-theoretic interpretation to choosing KL divergence as the distance function. Finally, several recent works have investigated the problem of learning high-dimensional distributions in this stronger KL divergence guarantee, such as [12, 32, 24, 11, 80].

## 3 Our Techniques

**Online Learning Framework to Learning in reverse KL**  Given i.i.d. samples from a distribution $P^*$, we are trying to learn it. Roughly, for a random $x \sim P^*$, a good approximate Bayes net $P$ should maximize the probability $P(x)$, or equivalently, minimize the expected log-likelihood $\mathbb{E}_{x \sim P^*}\left[\log \frac{1}{P(x)}\right]$. Keeping this in mind, we define the loss of any Bayes net $\widehat{P}$ predicted by the algorithm to be $\log \frac{1}{\widehat{P}(x)}$ for a sample $x$.

We follow the online learning framework. Here the algorithm $\mathcal{A}$ observes a set of samples $x^{(1)}, \ldots, x^{(T)} \sim P^*$ over $T$ rounds from an unknown Bayes net $P^*$. The goal of the algorithm is to learn a distribution $\widehat{P}$ which is close to $P^*$. After observing each sample $x^{(t)}$, $\mathcal{A}$ predicts a Bayes net $\widehat{P}_t$ and incurs a loss of $\log \frac{1}{\widehat{P}_t(x^{(t)})}$ for this round. However, there is a set of experts $\mathcal{E}$ to help $\mathcal{A}$. For simplicity, we can assume each expert $E \in \mathcal{E}$ is simply one Bayes net among all possible Bayes nets. Had $\mathcal{A}$ stuck to any particular expert $E \in \mathcal{E}$, it would incur a total loss $\sum_{t=1}^{T} \log \frac{1}{E(x^{(t)})}$ over all the $T$ rounds. The algorithm can change the experts in between or do some randomized strategy for choosing the expert among $\mathcal{E}$. Let $\widehat{P}_t$ be its prediction after round $t$. The regret is defined to be the difference between the loss of the algorithm and that of the best expert: $\sum_{t=1}^{T} \log \frac{1}{E(x^{(t)})} - \min_{E \in \mathcal{E}} \sum_{t=1}^{T} \log \frac{1}{E(x^{(t)})}$.

In our setting, the expert set consists of one Bayes net per DAG from the class of DAGs under consideration (e.g., acyclic orientations of a given skeleton). To associate a Bayes net with a DAG, we approximately learn the conditional distributions at each node using the *add-one*  or *Laplace* estimator on a separate set of samples. We have the guarantee that these finitely many Bayes nets form a "cover" for the class of Bayes nets we wish to learn.

We relate the regret mentioned above to the expected average of the KL divergence over the rounds: $\mathbb{E}[\frac{1}{T}\mathsf{D}_{\mathsf{KL}}(P^*||\widehat{P}_t)]$. Once we control the average KL divergence, using the convexity of KL, we can show that the mixture distribution $\frac{1}{T}\sum_{t=1}^{T}\widehat{P}_t$ is also close to $P^*$. Finally, we translate the above bounds from expectation to high probability using McDiarmid's bounded difference inequality.

We use known bounds on the regret for two classic online learning algorithms: the Exponential Weighted Average (EWA) algorithm and the Randomized Weighted Majority (RWM) algorithm. EWA returns us the mixture $\frac{1}{T}\sum_{t=1}^{T}\widehat{P}_t$ which improperly learns $P^*$ in (reverse) KL. RWM returns a

random Bayes net $\widehat{P}$ which properly learns $P^*$ in expected KL. The pseudocode for these algorithms is given in Algorithm 1 and Algorithm 2.

---

**Algorithm 1:** EWA-based learning for Bayes nets

---

**Input** : $\mathcal{N} = \{P_1, \ldots, P_N\}, T,$ hyperparameter $\eta > 0$.
**Output :** Sampler for $\widehat{P}$.

1 $w_{i,0} \leftarrow 1$ for each $i \in [N]$.
2 **for** $t \leftarrow 1$ *to* $T$ **do**
3      Observe sample $x^{(t)} \sim P^*$.
4      Update $w_{i,t} \leftarrow w_{i,t-1} \cdot P_i(x^{(t)})^\eta$ for each $i \in [N]$.
5 **function** EWA-SAMPLER()
6      Sample $t \leftarrow [T]$ uniformly, then sample $i \sim [N]$ with probability $\frac{w_{i,t-1}}{\sum_{j \in [N]} w_{j,t-1}}$.
7      **return** $x \sim P_i$.
8 **return** EWA-SAMPLER /* This is a sampler for $\widehat{P}$. */

---

**Algorithm 2:** RWM-based learning for Bayes nets

---

**Input** : $\mathcal{N} = \{P_1, \ldots, P_N\}, T,$ hyperparameter $\eta > 0$.
**Output :** $\widehat{P} \in \mathcal{N}$.

1 $w_{i,0} \leftarrow 1$ for each $i \in [N]$.
2 **for** $t \leftarrow 1$ *to* $T$ **do**
3      Sample $i_t$ from $[N]$ with $\Pr(i_t = i) = \frac{w_{i,t-1}}{\sum_{j \in [N]} w_{j,t-1}}$.
4      Observe sample $x^{(t)} \sim P^*$.
5      **for** $i \in [N]$ **do**
6          $w_{i,t} \leftarrow w_{i,t-1} \cdot P_i(x^{(t)})^\eta$.
7 Sample $t$ uniformly from $[T]$.
8 **return** $\widehat{P} \leftarrow P_{i_t}$.

---

**Efficient Learning of Restricted Classes of Bayes Nets** Our learning algorithm for Bayes nets mentioned above is sample-optimal but not time-efficient in general since the number of experts to be maintained is of exponential size. However, we observe that for special cases of Bayes nets, we can efficiently sample from the experts according to the randomized strategy of the algorithm. As a remark, the idea that the computational barrier of RWM or EWA may be side-stepped by developing efficient sampling schemes was also used in a recent work on fast equilibrium computation in structured games [8] and partly motivated our work.

To see the simplest example of this idea, suppose $\mathcal{P} = \{P_1, \ldots, P_N\}$ is a set of distributions over $[k]$, and let $\mathcal{P}^{\otimes n} = \mathcal{P} \times \mathcal{P} \times \cdots \times \mathcal{P}$ be a set of product distributions over $[k]^n$. Each element of $\mathcal{P}^{\otimes n}$ is indexed as $P_{\mathbf{i}}$ for $\mathbf{i} = (i_1, \ldots, i_n)$, so that $P_{\mathbf{i}}(x) = \prod_{j=1}^n P_{i_j}(x_j)$. The size of $\mathcal{P}^{\otimes n}$ is clearly $N^n$, so it is infeasible to work with it directly. The RWM algorithm maintains a distribution over $\mathcal{P}^{\otimes n}$, so that the probability that RWM picks $P_{\mathbf{i}}$ for its prediction $\widehat{P}_t$ at time $t$ is proportional to $\prod_{s=1}^{t-1} P_{\mathbf{i}}(x^{(s)})^\eta$, where $x^{(s)}$ is the observed sample at time $s$ and $\eta > 0$ is a parameter. Therefore:

$$\Pr_{\mathsf{RWM}}[\widehat{P}_t = P_{\mathbf{i}}] = \frac{\prod_{s=1}^{t-1} P_{\mathbf{i}}(x^{(s)})^\eta}{\sum_{\mathbf{i'}} \prod_{s=1}^{t-1} P_{\mathbf{i'}}(x^{(s)})^\eta} = \prod_{j=1}^n \frac{\prod_{s=1}^{t-1} P_{i_j}(x_j^{(s)})^\eta}{\sum_{i_j'} \prod_{s=1}^{t-1} P_{i_j'}(x_j^{(s)})^\eta}.$$

The crucial observation is that RWM maintains a product distribution over product distributions, and so we can sample each $P_{i_j}$ from $\mathcal{P}$ independently.

When the underlying Bayes net is a tree, i.e. of indegree 1, we show that RWM samples a random rooted spanning arborescence from a weighted complete graph. The probability to output a particular arborescence $A$ is proportional to $\prod_{e \in A} w_e$ where each $w_e$ is a weight that can be explicitly computed in terms of the observed samples and the parameters of the algorithm. It is well-known that the *matrix-tree theorem* (more precisely, *Tutte's theorem*) for counting weighted arborescences can be used for this purpose, and hence, we obtain an alternative to the Chow-Liu algorithm for approximately learning a tree Bayes net efficiently and sample-optimally.

Next, we generalize our algorithm to polytree-structured Bayes nets where the underlying skeleton is acyclic. Here, we are assuming that the skeleton is given, so that the goal of the algorithm is to learn an acyclic orientation of the skeleton. For simplicity, suppose the skeleton is known to be the path. Given a particular orientation of the edges, we obtain a particular Bayes net structure. Once the structure is fixed, the conditional probability distribution corresponding to each edge parent→child is set according to the empirical statistics in a separate batch of samples. This will completely specify a Bayes net $P$, which can assign probability $P(x^{(t)})$ for the sample $x^{(t)}$. Therefore, we can also compute the total loss $\ell_P = \sum_{t=1}^T \log P(x^{(t)})^{-1}$. Then, each structure $P$ will be chosen proportional

to $e^{-\eta \ell_P}$ in the RWM algorithm. In order to sample a particular Bayes net among the entire class using RWM, we need to first compute the normalization constant of the RWM sampler's distribution: $Z := \sum_{P \in \mathcal{P}} e^{-\eta \ell_P}$ over the class $\mathcal{P}$ of all (discretized) path Bayes nets. A particular path Bayes net $P$ will be chosen with probability $e^{-\eta \ell_P}/Z$ by the RWM's sampler. We first show how to compute $Z$ efficiently using dynamic programming.

We now show how to compute $Z$ by induction on the set of vertices of the path. Suppose $Z_j$ is the normalization constant obtained by only restricting to the first $j + 1$ nodes in the path. That is, if $\mathcal{P}_j$ is the class of Bayes nets corresponding to all orientations of the undirected path on $j + 1$ nodes, then $Z_j = \sum_{P \in \mathcal{P}_j} e^{-\eta \ell_P}$, where $\ell_P$ only computes the loss based on the first $j + 1$ variables. For the induction, we maintain more refined information for each $j$. Let $\mathcal{P}_{j,\leftarrow}$ and $\mathcal{P}_{j,\rightarrow}$ be the class of all discretized Bayes nets on $j + 1$ variables with a path skeleton and the last edge pointing left and right, respectively. Correspondingly, define $Z_{j,\leftarrow}$ and $Z_{j,\rightarrow}$; clearly, $Z_j = Z_{j,\leftarrow} + Z_{j,\rightarrow}$. Inductively, assume that $Z_{j,\leftarrow}$ and $Z_{j,\rightarrow}$ are already computed. We then need to compute $Z_{j+1,\leftarrow}$ and $Z_{j+1,\rightarrow}$.

If the $(j+1)$-th edge orients rightward, then the parents of nodes $1, \ldots, j+1$ do not change, while the new node $j + 2$ has parent $j + 1$. We can accommodate this new edge by simply adding the negative log of the conditional probability due to this new edge to the loss restricted to the first $j + 1$ variables. We can compute $Z_{j+1,\rightarrow} = (Z_{j,\leftarrow} + Z_{j,\rightarrow})e^{-\eta \Delta}$, by computing $\Delta = \sum_{t=1}^{T} \log P(x_{j+2}^{(t)} \mid x_{j+1}^{(t)})^{-1}$.

If the $(j + 1)$-th edge orients leftward, the adjustment is slightly trickier as node $j + 1$ will get a new parent $j + 2$, while the new node $j + 2$ has no parent. In that case, we need to first subtract out the previous sum of negative log conditional probabilities at $j + 1$. Let us define:

$$\Delta_1 = \sum_{t=1}^{T} \log P(x_{j+1}^{(t)} \mid x_j^{(t)})^{-1}, \; \Delta_2 = \sum_{t=1}^{T} \log P(x_{j+1}^{(t)})^{-1}, \; \Delta_3 = \sum_{t=1}^{T} \log P(x_{j+1}^{(t)} \mid x_j^{(t)}, x_{j+2}^{(t)})^{-1},$$

$$\Delta_4 = \sum_{t=1}^{T} \log P(x_{j+1}^{(t)} \mid x_{j+2}^{(t)})^{-1}, \Delta_5 = \sum_{t=1}^{T} \log P(x_{j+2}^{(t)})^{-1}.$$

If node $j$ is not a parent of node $j + 1$, then node $j + 1$ contributed $\Delta_2$ loss to $Z_{j,\leftarrow}$ while now it contributes $\Delta_4$ loss to $Z_{j+1,\leftarrow}$. Otherwise, it contributed $\Delta_1$ loss to $Z_{j,\rightarrow}$ while now it contributes $\Delta_3$ loss to $Z_{j+1,\leftarrow}$. The new node $j + 2$ contributes $\Delta_5$ loss to $Z_{j+1,\leftarrow}$ independent of what happens to the other variables. Summarizing:

$$Z_{j+1,\leftarrow} = Z_{j,\leftarrow} \cdot e^{-\eta(\Delta_4 - \Delta_2 + \Delta_5)} + Z_{j,\rightarrow} \cdot e^{-\eta(\Delta_3 - \Delta_1 + \Delta_5)}.$$

It is easy to see that these updates can be performed efficiently using an appropriate dynamic programming table. Once we have computed the total sum $Z = Z_{n,\leftarrow} + Z_{n,\rightarrow}$, sampling a structure according to the sampler's distribution can simply be done by suitably unrolling the DP table.

Figure 1: Given a rooted polytree skeleton, for each node $v$, and for each fixed orientation of edges incident to $v$, we maintain the total weight of all consistent orientations of the subtree rooted at $v$. Above, the orientations of edges incident to $B$ and $C$ are fixed. This is needed when computing the weight for the subtree rooted at $A$, since in the first two panels, the in-degree of $C$ change from 1 to 2, while in the second two panels, $C$'s in-degree does not change.

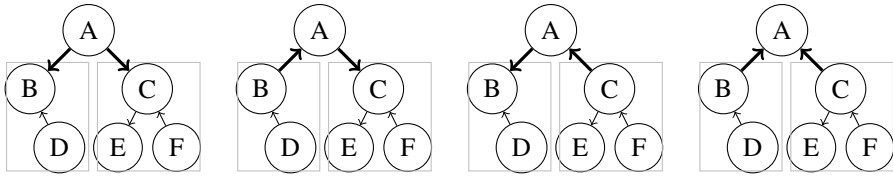

The argument described above extends to learning bounded indegree polytrees and bounded indegree chordal graphs. For polytrees, the idea is illustrated in Figure 1. For chordal graphs, the algorithm first builds a clique tree decomposition and uses this structure for dynamic programming. The obvious issue with chordal graphs is that some orientations may lead to cycles, unlike the case for polytrees. However, chordal graphs enjoy certain nice property (see Lemma C.6) that allows us to independently perform weighted counting/sampling of acyclic orientations in each subtree of the clique tree.

**Agnostic Learning via Maximum Likelihood Estimation** An arguably more natural approach to PAC learning in KL divergence is to maximize the empirical log-likelihood (MLE) over a suitably-discretized class of distributions (e.g., see [41], Theorem 17). The Chow-Liu algorithm for tree distributions can also be viewed through this lens. Note however that, despite a long history of study, Chow-Liu is not known to attain the sample complexity bound in theorem 2.4 for learning trees in the realizable setting.

For the problem of learning polytrees and chordal-structured distributions, we can in fact adapt our algorithm to maximize likelihood and thus, get a sample complexity bound which is comparable to our Theorem B.12 (up to log factors) for proper learning in KL. But it does not yield the near-optimal bounds (for constant failure probability) that we get for improper learning (Theorem B.11) in the realizable case. The challenge of implementing the Maximum Likelihood (ML) algorithm over an exponential-sized class of distributions is *efficiency* — a naive approach would take exponential time. The dynamic programming algorithms that we develop for efficient weighted counting and sampling of DAG structures (which we use to implement EWA / RWM) can also be used to implement MLE efficiently for polytree/chordal-structured distributions given the skeleton. We give an outline of this in Appendix G.

## 4 Related Works

[51] gave the first sample complexity bounds for agnostic learning of a Bayes net with known structure from samples in KL divergence. This work also gave an efficient algorithm for special cases such as trees using the classical Chow-Liu algorithm. Subsequently, [28] gave an efficient algorithm for learning an unknown Bayes net (discrete and Gaussian) on a fixed structure. This result was improved to a sample-optimal learning of fixed-structure Bayes nets in [10, 12].

The general problem of distribution learning of Bayes networks with unknown DAG structure has remained elusive so far. It has not been shown to be NP-hard, although some related problems and specific approaches are NP-hard [21, 23, 26, 55]. Many of the early approaches required *faithfulness*, a condition which permits learning of the Markov equivalence class, e.g. [73, 22, 47]. Finite sample complexity of such algorithms has also been studied, e.g. [45]. Specifically for polytrees, [68, 49] studied recovery of the DAG for polytrees under the infinite sample regime and in the realizable setting, while [24] gave finite sample complexity for this problem. [48] studied the more general problem of learning Bayes nets, and their sufficient conditions simplified in the setting of polytrees.

A notable prior work in the context of the current paper is the work by [1], which also explores the improper learning of Bayesian networks with polynomial sample and time complexities. However, our research diverges from theirs in three critical ways: firstly, their study does not offer any proper learning algorithms; secondly, it lacks agnostic learning guarantees; and thirdly, their approach does not achieve optimal sample complexity in the realizable setting. While they do demonstrate a "graceful degradation" as inputs deviate from the hypothesis class, this does not equate to a definitive agnostic learning guarantee as provided in our work. On a positive note, their research does attain polynomial sample and time complexities for learning any Bayesian network with a bounded total degree in the realizable setting. It is worth noting that our results for distributions with chordal skeleton are applicable even when the total degree is unbounded, provided that the indegree remains bounded, a scenario where the findings of Abbeel, Koller, and Ng would not be applicable.

**Online Learning of Structured Distributions** The approach of using the online learning framework for distribution learning has been considered in the literature [18, 83, 77]. These works use EWA algorithm and output the mixture distribution. However, they primarily focus on minimizing the number of samples, and are not computationally efficient in general. Since we are interested in computationally efficient learning of high-dimensional distributions, their approaches do not directly translate to our context. The closest we get is the *Sparsitron* algorithm by Klivans and Meka ([57]) which learns an unknown Ising model from samples. However, Sparsitron is typical to Ising models where the conditional distribution at any component follows a logistic regression model which the Sparsitron algorithm learns.

Although not for distribution learning, a similar use of the multiplicative weights update method appears in Freund and Schapire's well-known AdaBoost algorithm ([43]) where the algorithm

---

[3]This work studies a more general notion of factor graphs.

| | Structure | Efficient? | Agnostic? | Additional assumptions |
|---|---|---|---|---|
| [12] | Tree | Yes | Yes | None |
| [24] | Polytree | Yes | No | Known skeleton |
| [1] | Bounded total degree [3] | Yes | No | None |
| [14] | Bounded in-degree | No | No | None |
| Our results | Tree | Yes | Yes | None |
| | Chordal skeleton, bounded in-degree | Yes | Yes | Known skeleton |

Table 2: Comparison with existing works.

implicitly creates a sequence of probability measures. Later work on the hard-core lemma, such as [7], explicitly focus on efficient sampling from the iterates of multiplicative weights update.

**Robust Learning** In the field of distribution learning, it is commonly assumed that all samples are consistently coming from an unknown distribution. However, real-world conditions often challenge this assumption, as samples may become corrupted—either removed or substituted by samples from an alternate distribution. Under such circumstances, the theoretical assurances of traditional algorithms may no longer apply. This discrepancy has spurred interest in developing robust learning algorithms capable of tolerating sample corruption. Recent years have seen notable advancements in this area, including the development of algorithms for robustly learning Bernoulli product distributions [37], and enhancing the robustness of learning Bayes nets [20]. See [62, 34, 35, 6, 52, 59, 35, 36, 19, 15] and the references therein for a sample of current works in this area. These works primarily focus on guarantees with respect to the total variation distance.

Of particular relevance is the TV-*contamination model*. Here, if the distribution to be learnt is $P$, one gets samples from a 'contaminated' distribution $Q$ with $\mathsf{d}_{\mathsf{TV}}(P,Q) \leq \eta$. Note that this is a stronger model than *Huber contamination* ([54]), where the noise is restricted to be additive, meaning that an adversary adds a limited number of noisy points to a set of uncontaminated samples from $P$.

One can interpret our results using a KL-*contamination model*. If the distribution to be learnt is an unknown $P$ promised to belong to a class $\mathcal{C}$, the contaminated distribution $Q$ is some distribution satisfying $\mathsf{D}_{\mathsf{KL}}(Q\|P) \leq \eta$. The noise is again non-additive, but the model is weaker than TV-contamination. Any $(\eta, A)$ approximation for $Q$ with respect to $\mathcal{C}$ yields a distribution $\widehat{P}$ such that $\mathsf{D}_{\mathsf{KL}}(Q\|\widehat{P}) \leq (A+1)\eta$. Therefore, we get that for Hellinger distance:

$$\mathsf{H}(P, \widehat{P}) \leq \mathsf{H}(P, Q) + \mathsf{H}(\widehat{P}, Q) \leq \sqrt{\eta} + \sqrt{(A+1)\eta} \leq \sqrt{(2A+3)\eta}.$$

Similarly, one can also bound $\mathsf{d}_{\mathsf{TV}}(P, \widehat{P}) = O(\sqrt{\eta})$ for constant $A$. To the best of our knowledge, the KL-contamination model has not been explicitly considered before, but if one were to directly apply the results of [20] with the assumption that $\mathsf{D}_{\mathsf{KL}}(Q\|P) \leq \eta$, one would obtain a distribution $\widehat{P}$ such that $\mathsf{d}_{\mathsf{TV}}(P, \widehat{P}) = O(\sqrt{\eta \log 1/\eta})$, worse than ours by a $\sqrt{\log 1/\eta}$ factor which seems unavoidable using their approach [38]. Moreover, their results require that $\mathcal{C}$ be a class of *balanced* Bayes nets, a technical condition which is not needed for our analysis [4]. However, we would like to note that $\mathsf{D}_{\mathsf{KL}}(Q\|P)$ can be large as compared to $\mathsf{d}_{\mathsf{TV}}(Q, P)$, so this holds when $\mathsf{D}_{\mathsf{KL}}(Q\|P)$ is small.

## 5 Discussion

**Conclusion** In this work, we established a novel connection between distribution learning and graphical structure sampling algorithms via the framework of online learning. Leveraging this connection, we designed efficient algorithms for agnostically learning bounded indegree chordal-structured distributions, with polynomial sample complexity. These algorithms only require knowledge of the

---

[4] A Bayes net is said to be *c-balanced* for some $c > 0$ if all conditional probability table values $\in [c, 1-c]$.

distribution's skeleton, without needing information on the edge directions. Since polytree-structured distributions are a subset of chordal-structured distributions, our result also gives new results on the well-studied problem of learning polytree-structured distributions. Interestingly, our method also leads to a new algorithm for learning tree-structured distributions, which is significantly different from the extremely well studied Chow-Liu algorithm. Finally, we also give an improper learning algorithm that, with probability $2/3$, gives an $(\varepsilon, 3)$-approximation with respect to tree-structured distributions, which has a quadratic sample complexity advantage over Chow-Liu.

**Organization of the supplementary material**  Due to shortage of space, the rest of the paper is presented in the supplementary material. It is organized as follows. In Appendix A, we present the preliminaries required for this work. Appendix B establishes the connection between regret in online learning to KL divergence in the scenario of agnostic learning of distributions. It also presents several necessary techniques from online learning along with the EWA and RWM algorithms that will be used later in our work. In Appendix C, we present our results on learning chordal-structured distributions. In Appendix D, we discuss our results on learning tree-structured distributions and present our alternative proper learning algorithm. In Appendix E, we give the lower bound of learning tree-structured distributions. In Appendix F, we design efficient learning algorithms for Bayes nets over graphs with bounded vertex cover. Finally, in Appendix G, we outline how our algorithms can be adapted to efficiently compute maximum likelihood.

## 6  Open Problems

Our work opens up several interesting research avenues.

- An intriguing question is whether we can extend our result for chordal graphs of bounded indegree to general graphs of bounded treewidth and bounded indegree. Interestingly, [74] showed that counting the number of acyclic orientations reduces to the evaluation of the Tutte polynomial at the point (2,0), and the Tutte polynomial can be evaluated efficiently for bounded treewidth graphs [65, 5]. This is relevant because the weights that EWA/RWM maintain are in some sense a weighted count of the number of acyclic orientations of the skeleton. However, we did not find a deletion-contraction recurrence for these weights, and so their connection to the Tutte polynomial is unclear.

- Another important follow-up direction for learning Bayes nets would be to search over *Markov equivalence classes* rather than DAG's. A Markov equivalence class corresponds to the set of DAGs that represent the same class of Bayes nets, and they can be represented as partially directed graphs (*essential graphs*) that satisfy some special graphical properties. It would be interesting to explore if the structure of essential graphs can be used to speed up weighted counting and sampling; indeed, a very recent work by [71] gives a polynomial time algorithm for uniformly sampling an essential graph that is consistent with a given skeleton.

- What is the role of *approximate sampling* in the context of distribution learning? So far, in this work, we have only used exact sampling algorithms for spanning arborescences and acyclic orientations. Can Markov chain techniques be brought to good use here? Our work further motivates settling the complexity status of approximately counting the number of acyclic orientations of an undirected graph; this question is a long-standing open problem in the counting/sampling literature.

- Finally, while we have restricted ourselves to learning Bayes nets here, our framework is quite general and also applies to learning other classes of distributions, such as Ising models and factor models. We leave these questions for future work.

## Acknowledgments and Disclosure of Funding

The authors would like to thank the anonymous reviewers for their comments which improved the presentation of the paper. AB and PGJ's research were supported by the National Research Foundation, Prime Minister's Office, Singapore under its Campus for Research Excellence and Technological Enterprise (CREATE) programme. SS's research is supported by the NRF Investigatorship award (NRF-NRFI10-2024-0006) and CQT Young Researcher Career Development Grant (25-YRCDG-SS).

AB and SS were also supported by National Research Foundation Singapore under its NRF Fellowship Programme (NRF-NRFFAI1-2019-0002). AB was additionally supported by an Amazon Research Award and a Google South/Southeast Asia Research Award. SG's work is partially supported by the SERB CRG Award CRG/2022/007985. NVV's work was supported in part by NSF CCF grants 2130608 and 2342244 and a UNL Grand Challenges Catalyst Competition Grant.

We would like to thank Debojyoti Dey, a Ph.D. student at IIT Kanpur, for discussions regarding robust learning algorithms in high dimensions. AB would also like to thank Daniel Beaglehole for a short meeting which seeded the idea for this work.

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
