# OpenReview forum: "Distribution Learning Meets Graph Structure Sampling"
_NeurIPS.cc/2025/Conference — NeurIPS 2025 poster_

### Official Review · Reviewer_exoZ · 2025-06-10

**Clarity:** 2
**Significance:** 3
**Originality:** 3
**Rating:** 4
**Confidence:** 3

**Summary:**

This paper uses ideas from online learning to make progress on PAC learning of Bayes networks.
They present an efficient (agnostic) PAC learning algorithm when the Bayes networks has a known chordal skeleton and improve in several ways in the case where the graph is a known tree which was studied in recent work [11].

**Questions:**

While the high level picture is clear - I am not quite sure I see the argument but this may be due to space constraints.
In particular it would be good to at least hint at the body of the paper as to:
1. Where do you use the fact that the graph is chordal?
2. It is not clear how local estimates are sufficient to compute the probability of each particular orientation.
3. Moreover, given that you consider potentially unbounded degree graphs, it is not clear how much data do you need in order to estimate the local weights especially as these need to be very accurate to apply randomized weighted majority.

**Ethical Concerns:**

["NO or VERY MINOR ethics concerns only"]

**Final Justification:**

This is nice theoretical progress on a hard problem. It is not clear in which case the graphical model is known but not the orientation. Finding some neat application would make the paper stronger. Similarly hinting as to how the algorithm works in the body of the paper would make it more attractive if possible.

**Limitations:**

The main limitation is the sparsity of the arguments provided. There is just not enough to convince that it works out.

**Quality:**

3

**Strengths And Weaknesses:**

The idea of treating different orientations as different experts is definitely attractive.
Similarly improving the efficiency by using conditional independence and estimate conditional probabilities from samples is very natural.

Of course restricting to a known graphical structure where only the orientation is unknown is theoretically interesting but maybe hard to motivate from applications?

The ideas of the proofs are a bit too high level and I am not sure how confident I am in their correctness.

---

> ### Author Rebuttal · Authors · 2025-07-31
>
> Thank you for your review. We address your comments and questions below:
>
> **Use of the chordal property of graph:** In the chordal distribution learning algorithm (Algorithm 7 in Section C in supplementary material), we crucially use the clique tree decomposition of the skeleton of the chordal graph in the subroutine SamplingChordalDist (Algorithm 6). Moreover, the chordal structure assumption is used for the correctness proof of our algorithm (Lemma C.6)
>
> **Computing the probability of each particular orientation using local estimates:** We would like to note that we formally prove that local estimates (of the add-one distribution) are sufficient (Lemma B.4 and Lemma B.6 in Section B in supplementary material). This does rely on the tensorization properties of KL divergence etc.
>
> **Estimating the local weights:** We do formally prove the sample complexity bounds, taking the indegree as a parameter. The sample complexities are, of course, polynomial only when the indegree is a constant, but this is similar to the information-theoretic lower bounds to learn Bayesian network (Canonne et al., 2020, Appendix B), which are exponential in terms of indegree. Especially, our improper learning sample complexities for general match the lower bounds up to log-factors (for indegree $n-1$, exponential samples are required in any case since general distributions on $[k]^n$ can be represented by such Bayes nets).
>
> **Sparsity of the arguments:** Unfortunately, space constraints prevented us from including the arguments in the main paper.  We have included the complete arguments in the supplementary material.
>
> **Reference**
>
> Clément L Canonne, Ilias Diakonikolas, Daniel M Kane, and Alistair Stewart. Testing Bayesian Networks, IEEE Trans. Inf. Theory, 2020.

---

### Official Review · Reviewer_eruZ · 2025-07-02

**Clarity:** 2
**Significance:** 2
**Originality:** 2
**Rating:** 2
**Confidence:** 2

**Summary:**

This paper tackles the PAC-learning of Bayesian network distributions by reducing the problem to online expert learning over all DAGs with a given chordal skeleton. The authors propose treating each possible DAG as an expert, aggregating them via efficient combinatorial counting of acyclic orientations. They prove that for a chordal skeleton with max indegree d, their algorithm learns a distribution within ε in KL-divergence using $$Õ(n³k^{d+1}/ε^²)$$ samples, extending classic Chow–Liu tree and polytree results.

**Questions:**

1. **Skeleton acquisition:** How would one obtain a correct chordal skeleton in practice? Would skeleton estimation errors undermine the guarantees?
2. **Unknown structure:** Can the online framework be extended to learn the skeleton itself (beyond the tree case), or is that inherently intractable?
3. **Causal interpretation:** Please clarify the role of “latent confounders” in this setting—does your method handle hidden-variable models, or is it purely observational?
4. **Empirical feasibility:** Have you attempted small-scale simulations (e.g., n≈20) to evaluate sample complexity and runtime in practice?

**Ethical Concerns:**

["NO or VERY MINOR ethics concerns only"]

**Final Justification:**

This paper makes a good theoretical contribution by giving the first efficient agnostic PAC learner for bounded‐indegree chordal‐skeleton Bayes nets, extending known results for trees and polytrees. The online learning–based reduction to structure sampling is elegant and well‐proved. However, the reliance on a known skeleton, no analysis of robustness to skeleton errors, and no empirical validation limit the work’s applicability. Given these gaps, I lean towards weak reject to reject recommendation, despite appreciating the theoretical advances.

**Limitations:**

The authors do not experimentally assess limitations; a discussion of skeleton errors, runtime constants, and potential failure modes would be beneficial.

**Quality:**

3

**Strengths And Weaknesses:**

**Strengths**
- **Novel theoretical bridge:** Clever reduction of Bayes net learning to online no-regret expert aggregation using combinatorial graph algorithms.
- **Generalization:** First efficient agnostic PAC learner for chordal-skeleton Bayes nets, subsuming tree and polytree cases.
- **Rigorous bounds:** Provable sample complexity and runtime guarantees grounded in well-established online learning theory.

**Weaknesses**
- **Known skeleton assumption:** Sidesteps the hardest part—structure discovery—raising questions about practical usability when the skeleton is unknown or estimated.
- **No experiments:** Lacks any empirical validation or simulations to gauge performance, leaving constants and scalability unclear.
- **Misleading framing:** Title mentions “latent confounder” and causality, but the work focuses purely on distribution learning under a known graph, which may confuse readers.

---

> ### Author Rebuttal · Authors · 2025-07-31
>
> Thank you for your review. We address your comments and questions below:
>
> **Known skeleton assumption:** We can still combine our algorithm with a practical structure learning algorithm, as noted in our reply to Reviewer c9e7.  If the structure learning algorithm has theoretical guarantee that it ``exactly’’ recovers the structure, we can get a theoretical guarantee for this combined algorithm. Otherwise, this does not give the proper theoretical guarantee with respect to the true distribution. Unfortunately, to the best of our knowledge, we do not know of such an efficient structure learning algorithm with exact recovery guarantees.
>
> **Misleading framing:** We are afraid that we do not know why the reviewer makes such a comment. Our title (“Distribution Learning Meets Graph Structure Sampling”) clearly does not mention “latent confounder” or “causality”. Furthermore, we do not mention “latent confounders” anywhere in our paper, and the only place where we mention “causality” is lines 99-100 where we say that chordal graphs play a crucial role in the study of causal Bayesian networks. We stand by this statement and we do not feel it is misleading in any way.
>
> **Skeleton acquisition:** There have been existing works on practical structured learning algorithms (cite survey by Daly et al., 2011). Unfortunately, these do not often have theoretical guarantees for “exact recovery”. If we combine a structure learning algorithm with our distribution learning approach, it would give us the proper guarantee as we state in the paper (without the skeleton assumption) if the correct skeleton is output by the structure learning algorithm. If the structure learning phase outputs a wrong skeleton $G_{err}$, we can get an agnostic learning guarantee with respect to the set of distributions with that structure $G_{err}$ (not wrt the true skeleton $G$). The KL divergence (from the output distribution to the true distribution) in this case may be high, even for “structurally-close” skeletons (e.g., if the learnt skeleton $G_{err}$ is missing a single edge from $G$).
>
> **Unknown structure:** The general framework in Section B in the supplementary material does not rely on having a known skeleton, and neither does the online-learning based tree-structured distribution learning algorithm in Section D in the supplementary material. In that sense, the online learning framework can indeed be applied to efficiently learn the skeleton as well as the probabilities without knowing the skeleton in advance. But this efficient algorithm does rely on a strong combinatorics result (the generalization of the matrix-tree theorem, Lemma D.13 in the supplementary material), which applies only to rooted trees as far as we know. We do not have a general computationally-efficient algorithm that goes over all possible skeletons for polytrees/chordal graphs.
>
> **Causal interpretation:** Unfortunately, our work does not mention “latent confounders” anywhere in the main paper, as well as in the supplementary material.
>
> **Empirical feasibility:** In this work, we focused on proving sound theoretical guarantees of our results, which will continue to hold in practice. We defer the practical implementation of our results for future work.
>
> **Reference:**
>
> Daly, R, Qiang, S & Aitken, S, Learning Bayesian Networks: Approaches and Issues, Knowledge Engineering Review, 2011.

---

### Official Review · Reviewer_c9e7 · 2025-07-02

**Clarity:** 2
**Significance:** 4
**Originality:** 3
**Rating:** 5
**Confidence:** 1

**Summary:**

This paper proposes a novel distribution learning algorithm for graphical models. The proposed method is the first to enable PAC learning of bounded indegree chordal-structured distributions using only the skeleton of the graphical model as prior information. It can also be applied to learning polytree-structured and tree-structured distributions, and is shown to outperform existing methods theoretically. The key idea behind the proposed approach is to leverage algorithms from the online learning literature, such as the Exponential Weighted Average algorithm and the Randomized Weighted Majority algorithm. By exploiting structural properties of graphical models, the method enables efficient sampling of experts within the algorithm.

**Questions:**

- While the proposed method relies on several assumptions—such as the distribution being a bounded indegree chordal-structured distribution and the skeleton being known—can these limitations be addressed or relaxed through extensions of the proposed framework?
- In the context of practical applications, to what extent do the assumptions made by the proposed method (e.g., known skeleton, chordal structure) pose a barrier? Are they relatively benign, or could they significantly hinder real-world deployment?

**Ethical Concerns:**

["NO or VERY MINOR ethics concerns only"]

**Final Justification:**

My concerns have been resolved. Since I had originally given a score leaning toward acceptance, I will keep the score as it is.

**Limitations:**

The proposed method makes various assumptions about the shape and input of graphical models, limiting the problems to which it can be applied.

**Quality:**

3

**Strengths And Weaknesses:**

As I am not an expert in the learning theory of graphical models, I may not be able to accurately assess the validity or correctness of the authors’ claims. I have not wholly followed all the proofs provided in the appendix.

# Strengths

- The proposed method is innovative and elegant. It applies techniques from online learning to estimating distributions over graphical problems and suggests promising directions for future research.
- The method is extensively analyzed from a theoretical perspective, and improvements over existing approaches are clearly demonstrated.

# Weaknesses

- Certain assumptions limit the applicability of the proposed method, such as the requirement that knowledge of the graphical model's skeleton and the distribution's chordal structure.
- The paper has a complex structure, making it difficult to follow. Understanding the overall framework requires reading the appendix in detail. While key components necessary for understanding the method are deferred to the appendix, some arguably less critical content (e.g., the paragraph "Why KL divergence?") remains in the main text. The organization between the main body and the appendix warrants reconsideration.

---

> ### Author Rebuttal · Authors · 2025-07-31
>
> Thank you for your review. We address your comments and questions below:
>
> **Assumptions about the knowledge of the graphical model's skeleton and the distribution's chordal structure on the proposed method:** The general framework presented in Section B of the supplementary material does not rely on these assumptions. Of course, the weakness is that the meta-algorithms (Algorithms 3 and 4 in Section B) would take exponential time (even with good sample complexity) and are thus not useful in practice. The tree-structured distribution assumption and the known chordal skeleton assumption are used in the algorithms developed in Sections C and D (Algorithm 8 and Algorithm 7 in the supplementary material) to make the learning process computationally efficient. Note that the sample complexity as well as the computational efficiency of learning Bayes nets in this framework is tied to the indegree bound. This is because without an indegree bound or some other bound on the number of parameters, the sample complexity of learning Bayes nets would be exponential since they can represent general distributions of exponential size.
>
> **Complex structure:** Thanks for your suggestion. Unfortunately, due to space constraints in the main paper, we came up with this organization of the paper. We will make suitable changes in the final version.
>
> **Assumptions of the proposed method in the context of practical applications:** The algorithm in practice does not rely on the known skeleton assumption (of course, it is required in the analysis). Any structure learning algorithm (not necessarily with theoretical guarantees) that gives a chordal skeleton (with bounded indegree) $G_{err}$ as output (when $G$ is input) can be combined with our distribution learning algorithm to give an efficiently-samplable distribution as the output. However, the agnostic learning guarantee would only hold with respect to the set of distributions with structure $G_{err}$ rather than the true skeleton $G$. On the other hand, the chordal-structure and bounded-indegree assumptions are fundamentally required for our computationally-efficient algorithm.

---

> > ### Comment · Reviewer_c9e7 · 2025-08-05
> >
> > Thank you for the rebuttal; it has fully resolved my questions. So, I will keep my current score.
> > As noted in my review, I am not an expert on the work’s novelty or significance; thus, my confidence in those evaluations is limited.

---

### Official Review · Reviewer_aMQD · 2025-07-02

**Clarity:** 4
**Significance:** 3
**Originality:** 3
**Rating:** 4
**Confidence:** 4

**Summary:**

This paper gives computationally efficient on-line learning algorithms for certain families of Bayesian Networks, models with a known skeleton with a chordal graph of bounded in-degree, and trees. A computationally efficient algorithm for trees was known (indeed, the algorithm itself, by Chow and Liu, is very old) but the proposed method obtains optimal sample complexity -- the dependence on the size of the range of the random variables is improved to quadratic, rather than cubic. The method yields guarantees for agnostic improper learning of such structures. (A further extension to learning Bayes Nets with a moralization with a O(1)-size vertex cover, using another prior work for sampling, is described in an appendix.)

These results are obtained by leveraging the following simple observation: the multiplicative updates used in existing on-line learning methods can be distributed across the factors of the probabilistic model. This is essentially equivalent to running a separate learner for the conditional probability tables for each of the factors; this corresponds to a mixture of models that can be sampled by independently sampling a table for each factor.

**Questions:**

Do you have a deeper link between counting/sampling of graph structures and learning graphical models? (Besides the application here, and the casting of the computation of partition functions as a counting problem; I mean, is there a tight link between these problems, as the abstract and introduction seems to suggest? Where is it spelled out in the paper?)

**Ethical Concerns:**

["NO or VERY MINOR ethics concerns only"]

**Final Justification:**

My recommendation remains more or less unchanged:

On the positive side, there is still a cute theoretical idea in the maintenance of the weights node-wise that yields a polynomial time algorithm where one was previously not known.

But, on the negative side, the discussion with the authors seems to confirm the main weaknesses: (1) that the algorithm is still wildly impractical on account of the number of discretized conditional probability tables, each of which must maintain a weight per node, as well as (2) that the "connection" to sampling methods is, rather, just an application and (3) the paper only gives methods for fitting the local tables and edge orientations, not learning the structure.

The main idea has merit, and there is something to learn from the paper, but the weaknesses temper my enthusiasm. I am not sure about the impact. Hence, "weak accept".

**Limitations:**

Yes

**Quality:**

3

**Strengths And Weaknesses:**

The paper has two main strengths. The first is that the underlying idea is very clean and well-suited to these kinds of probabilistic models. The second is that it indeed obtains quantitatively improved guarantees for specific problems, like learning tree-structured distributions.

The main weaknesses are first, that the approach relies on running the learner on a discretization of the conditional probability tables. Although the conditional probability tables themselves are inherently exponential in the number of assignments to the parents, and such a dependence is to be expected, the requirement to maintain parameters for the individual table values seems somewhat impractical. Second, the chordal graph learner relies on possession of a known skeleton for the DAG, and this approach doesn't address the structure learning problem that poses the more fundamental barrier to the use of probabilistic graphical models in practice. It is a theoretical improvement -- I am not aware of a method that is guaranteed to learn such structures -- but it is not clear if it makes a difference in practice.

The "novel link" between learning graphical models and sampling structures, described in the abstract, seems like an over-sell. The work uses sampling methods for improper learning of the models, but this is (just) an application, not the discovery of a deeper connection between the problems. (For example, the paper does not seem to establish a connection in the other direction.)

---

> ### Author Rebuttal · Authors · 2025-07-31
>
> Thank you for your review. We address your comments and questions below:
>
> **Exponential size conditional probability table:** We would like to note that while the EWA/RWM-based algorithms implemented “as is” (as described in Section B of the supplementary material) would use exponential-size tables and take exponential time, the way we implement it for trees/bounded-indegree chordal graphs via sampling does not use exponential-size tables (in the chordal graph case, if we assume that the indegree is constant). Thus, our results on learning tree and bounded-indegree chordal and polytree distributions work in polynomial time, as opposed to exponential time if we had used an exponential-size table. The computational efficiency of our algorithms is the non-trivial contribution of our paper.
>
> **Known skeleton assumption:** This is indeed a part left open by our paper (in terms of getting formal guarantees for structure learning). In practice, any structure learning method might be used, and the agnostic learning guarantee will hold with respect to the closest Bayes net (to the input distribution) having that structure.
>
> **Novel link between learning graphical models and sampling structures:** In this work, we have used the online learning and recursive graph structure sampling framework to efficiently learn tree, polytree and chordal structured distributions in KL divergence. We note in Section 3 that we can replace the online learning framework with maximum likelihood estimation (sacrificing the near-optimal sample complexities for improper learning), but even then the graph structure sampling is crucial for getting computationally-efficient algorithms. Unfortunately, we are not aware of a connection in the other direction (graph structure sampling via distribution learning), which seems unlikely in general. It is possible that given a learnt Bayes-net distribution as a density oracle, it could be used to learn the graph structure of that distribution in some non-trivial way which differs from learning the structure through samples (which is NP hard as shown in Chickering (2004)).
>
> **Reference:**
> Max Chickering, David Heckerman, and Chris Meek. Large-sample learning of bayesian networks is NP-hard. JMLR, 2004.

---

> > ### Comment · Reviewer_aMQD · 2025-08-02
> >
> > Regarding the conditional probability tables per node, the size is still exponential w.r.t. the in-degree (= number of parents), is it not? If not, two questions: (1) why do you require that the in-degree to be constant, and (2) how are you actually representing these conditional probabilities?

---

> > > ### Author Response · Authors · 2025-08-04
> > >
> > > Thank you for your question. Yes, the size of the conditional probability table for every node is exponential w.r.t. the in-degree d. In fact, any Bayes net with n nodes and indegree d requires $nk^{d+1}$ parameters. As a result, any polynomial (in $n,k, 1/\varepsilon$) sample learning algorithm needs to either bound the indegree or the number of parameters. Note that, without such a bound, Bayes nets on $[n]$ can represent general distributions over $n$-dimensional hypercube, which would require exponential in $n$ sample complexity for learning.
> > >
> > > For tree-structured distributions, as d=1, we obtain an algorithm with polynomial time and sample complexity. For chordal and polytree distributions, the sample complexity scales as follows: $\tilde{O}\left(\frac{n^4}{\varepsilon^4} + \frac{nk^{d+1}}{\varepsilon}\right)$ and $\tilde{O}\left(\frac{n^3}{\varepsilon^2 \delta^2} + \frac{nk^{d+1}}{\varepsilon}\right)$ for agnostic improper and proper learning, respectively, for alphabet size $k$, which becomes polynomial for constant $d$.

---

> > > > ### Comment · Reviewer_aMQD · 2025-08-05
> > > >
> > > > Let's put aside the sample complexity and talk about the computational complexity, which is the more relevant issue here. It looks like you have proposed to maintain weights for every value for the conditional probability table, is that correct? While I agree that for constant $d$ this is polynomial time, in order to attain a suitable accuracy overall the discretization parameter probably needs to scale with the number of variables, right? So it seems like you are maintaining something like $\sim (kn/\epsilon)^d$ weights per node, that you need to update on every round. Is this correct, or do you have a more computationally efficient way of maintaining the weights?

---

> > > > > ### Author Response · Authors · 2025-08-08
> > > > >
> > > > > Thank you for your comment. Let us clarify the runtime. At every node, for every choice of the d parents, we learn how the node conditionally depends on its parents. This is done at the outset, and after that, the algorithm focuses only on the combinatorial problem of learning the acyclic orientation. So, the runtime contribution from the node-distribution learning part is $\tilde{O}((\Delta k)^d \cdot n^2/\varepsilon)$, where $\Delta$ is the max degree of the skeleton, since $n \cdot {\Delta \choose d}$ bounds the number of all possible (node, parent set) pairings and $\tilde{O}(k^d n/\varepsilon)$ is the runtime of the add-1 algorithm at each node (Def B.7). Note that the runtime is polynomial even if both $\Delta$ and $d$ are $O(\log n)$.
> > > > >
> > > > > If $\Delta$ or $d$ are unbounded, then indeed the runtime can grow to be $n^d$. Again, note that for unbounded $d$, exponential dependence is inevitable, since degree-$n$ Bayes nets can capture arbitrary $n$-dimensional distributions.
> > > > >
> > > > > Thank you very much for the opportunity to reply. We will make sure to discuss this explicitly in the revision.

---

### Note · Authors · 2025-08-14

We thank all reviewers for their engagement which will inform improvements in the final version. They agreed  (exoZ: “definitely attractive”, eruZ: “clever reduction”, c9e7: “innovative and elegant”) about the novelty of our techniques. We revisit some of their concerns:

* _Unknown skeleton_: Conceptually, learning the topological order and learning the skeleton are separate issues, and there’s a long history of works separating the two; see [1,2,3].  The given skeleton needn’t be “correct”, as the output is near-optimal w.r.t. the given skeleton and so is robust in this sense to mis-specification. Our general framework doesn’t require the skeleton but has exponential runtime. It can be made efficient for trees and bounded vertex-cover graphs. We are optimistic that advances in counting algorithms can yield poly time algorithms for other graph classes.


* _Storage of conditional probability tables_: Reviewer aMQD is worried that we “maintain weights for every value for the conditional probability table”. We don’t. We fix a single conditional probability table, per node and choice of parent set. Having a degree bound is common in works on discrete Bayes nets (e.g., ref [13] in paper).


* _Lack of experiments_: We focus purely on rigorous sample and runtime bounds, since fundamental limits are still very much unknown. E.g, there’s no distribution learning hardness known for Bayes nets, especially via improper algorithms.


* _Organization_: Inevitably, the main paper with the 9 page limit only explains the ideas at a high level, but we believe they would still be interesting to a wider audience.

Reviewer eruZ writes: “Title mentions “latent confounder” and causality”; this must be the result of a confusion, as we don’t discuss these issues. Reviewer aMQD correctly points out that we only establish the connection between distribution learning and graph sampling in one direction. We will explicitly mention this, and incorporate the reviewers' other suggestions, in the revision.

[1] Bank, A., & Honorio, J. (2020). Provable efficient skeleton learning of encodable discrete bayes nets in poly-time and sample complexity. In ISIT.

[2] Shojaie, A., & Michailidis, G. (2010). Penalized likelihood methods for estimation of sparse high-dimensional directed acyclic graphs. Biometrika, 97(3), 519-538.

[3] Tsamardinos, I., Brown, L. E., & Aliferis, C. F. (2006). The max-min hill-climbing Bayesian network structure learning algorithm. Mach. Learn., 65(1), 31-78.

---

### Decision · Program_Chairs · 2025-09-17

**Decision:**

Accept (poster)

**Comment:**

This paper considers the task of learning directed graphical models with specific graphical structure. The main contribution is a method that leverages online learning to reduce learning to sampling certain high-dimensional distributions. This reduction is used to obtain improved sample complexity bounds and, in some cases, tractable computational complexity. With the exception of a single review with an unjustifiably low score, the other reviewers viewed this contribution as borderline with an inclination towards accepting the work. Given my own reading and understanding, I recommend borderline acceptance.